# Cartilage-Specific Gene Expression and Extracellular Matrix Deposition in the Course of Mesenchymal Stromal Cell Chondrogenic Differentiation in 3D Spheroid Culture

**DOI:** 10.3390/ijms25115695

**Published:** 2024-05-23

**Authors:** Igor V. Vakhrushev, Yulia B. Basok, Konstantin K. Baskaev, Victoria D. Novikova, Georgy E. Leonov, Alexey M. Grigoriev, Aleksandra D. Belova, Ludmila A. Kirsanova, Alexey Y. Lupatov, Veronika V. Burunova, Alexey V. Kovalev, Pavel I. Makarevich, Victor I. Sevastianov, Konstantin N. Yarygin

**Affiliations:** 1Laboratory of Cell Biology, Institute of Biomedical Chemistry, Moscow 119121, Russia; konstantinbaskaev@gmail.com (K.K.B.); vnlvikova@gmail.com (V.D.N.); golerus@gmail.com (G.E.L.); vburunova@mail.ru (V.V.B.); kyarygin@yandex.ru (K.N.Y.); 2Department for Biomedical Technologies and Tissue Engineering, Shumakov National Medical Research Center of Transplantology and Artificial Organs, Moscow 123182, Russia; bjb2005@mail.ru (Y.B.B.); bear-38@yandex.ru (A.M.G.); sashak1994@mail.ru (A.D.B.); lyudochkakirsanova@mail.ru (L.A.K.); viksev@yandex.ru (V.I.S.); 3Priorov Central Institute for Trauma and Orthopedics, Moscow 127299, Russia; kovalyov1@mail.ru; 4Institute for Regenerative Medicine, Medical Research and Education Centre, Lomonosov Moscow State University, Moscow 119192, Russia; pavel.makarevich@gmail.com; 5Institute of Biomedical Research and Technology, Moscow 123557, Russia

**Keywords:** mesenchymal stromal cells, chondrogenic differentiation, adipose tissue, umbilical cord, Wharton’s jelly, deciduous teeth, 3D cell culture, tissue spheroids, chondrospheres, cartilage regeneration

## Abstract

Articular cartilage damage still remains a major problem in orthopedical surgery. The development of tissue engineering techniques such as autologous chondrocyte implantation is a promising way to improve clinical outcomes. On the other hand, the clinical application of autologous chondrocytes has considerable limitations. Mesenchymal stromal cells (MSCs) from various tissues have been shown to possess chondrogenic differentiation potential, although to different degrees. In the present study, we assessed the alterations in chondrogenesis-related gene transcription rates and extracellular matrix deposition levels before and after the chondrogenic differentiation of MSCs in a 3D spheroid culture. MSCs were obtained from three different tissues: umbilical cord Wharton’s jelly (WJMSC—Wharton’s jelly mesenchymal stromal cells), adipose tissue (ATMSC—adipose tissue mesenchymal stromal cells), and the dental pulp of deciduous teeth (SHEDs—stem cells from human exfoliated deciduous teeth). Monolayer MSC cultures served as baseline controls. Newly formed 3D spheroids composed of MSCs previously grown in 2D cultures were precultured for 2 days in growth medium, and then, chondrogenic differentiation was induced by maintaining them in the TGF-β1-containing medium for 21 days. Among the MSC types studied, WJMSCs showed the most similarities with primary chondrocytes in terms of the upregulation of cartilage-specific gene expression. Interestingly, such upregulation occurred to some extent in all 3D spheroids, even prior to the addition of TGF-β1. These results confirm that the potential of Wharton’s jelly is on par with adipose tissue as a valuable cell source for cartilage engineering applications as well as for the treatment of osteoarthritis. The 3D spheroid environment on its own acts as a trigger for the chondrogenic differentiation of MSCs.

## 1. Introduction

Because of the lack of its own depot of progenitor cells and the absence of the microcirculatory system, the articular cartilage has a reduced ability to regenerate compared to most other tissues of the body. For this reason, cartilage defects often develop in the absence of discernible external damaging factors, just as a consequence of aging, repeated mechanical stresses, and metabolic disorders [1]. Over time, the articular cartilage wears down, leading to its decay, reactive inflammation, and further degradation. If untreated, this condition, named osteoarthritis, causes mobility limitation and severe pain syndrome, significantly reducing the patient’s quality of life.

There are many treatments for this condition, including orally taken medications (non-steroidal anti-inflammatory drugs, chondroprotective agents), injectables (glucocorticoids, hyaluronic acid), bone realignment, and joint replacement [2]. However, the traditional conservative methods for its treatment are aimed only at controlling symptoms and do not significantly affect the long-term prognosis, while surgery is complex and has many contraindications and side effects. Due to the rapid development of regenerative medicine in recent years, cellular and tissue-engineering technologies have become available, offering a unique opportunity to restore normal tissue in the area of injury. Initially, they were generally based on the implantation of the in vitro expanded autologous chondrocytes, either in suspension or adherent to or embedded in a biodegradable three-dimensional matrix, into the joint cavity [3,4]. The main disadvantage limiting the use of this approach is the need to obtain a sufficient number of the patient’s autologous chondrocytes, which is usually impossible in the case of the elderly, who are often suffering from degenerative diseases of the musculoskeletal system. Given the very low cell density in the articular cartilage, a large amount of biopsy material is required for effective sampling, which increases the risk of osteoarthritis occurrence at a new site. In addition to difficulties with the establishment of primary cultures from the biopsy material, cell expansion in vitro is also hampered because of the low mitotic index of chondrocytes and their predisposition to dedifferentiating into fibroblast-like cells during prolonged maintenance in the 2D culture. Other currently used autologous cell therapies include the intra-articular delivery of platelet-rich plasma (PRP) or the adipose tissue stromal vascular fraction (SVF) (reviewed in [5]).

Mesenchymal stromal (stem) cells (MSCs) probably have the potential to become a substitute for autologous chondrocytes because of their unique properties [6,7], such as their ability to undergo chondrogenic differentiation, high proliferative potential, availability, and immunosuppressive activity ensuring attenuation of chronic inflammation and the possibility of allogeneic transplantation. MSCs can be isolated from many tissues, such as bone marrow, subcutaneous adipose tissue, dental pulp, umbilical cord Wharton’s jelly, and many others. MSCs derived from different sources share a common CD phenotype, morphology, and other features [8,9]. However, depending on the tissue of origin, MSCs can differ significantly in their biological properties, including their propensity for directed differentiation [10,11]. Subcutaneous adipose tissue, umbilical cord Wharton’s jelly, and deciduous tooth pulp MSCs are among the common candidates for the role of starting material in cell therapy/tissue engineering translational research due to their availability and lack of ethical constraints. To date, there is still no consensus in the literature on which source of MSCs is optimal for cartilage regeneration.

Previously, using extracellular matrix (ECM) analysis, we obtained data supporting the hypothesis of the varying ability of MSCs from different tissues in chondrogenic differentiation [12]. The present study sought to conduct a more comprehensive comparison of the chondrogenic ability of these cells in 3D spheroid culture using chondrogenesis hallmarks, including the expression of cartilage-specific genes (Collagen I alpha 1 (COL1A1), Collagen II alpha 1 (COL2A1), Aggrecan (ACAN), and the sex-determining region Y (SRY)-box 9 (SOX9) [13,14], the synthesis of hyaline cartilage proteins, and ECM production. Primary chondrocytes served as the control. Changes in the level of gene expression in spheroid cells were assessed in comparison with cells in 2D culture.

## 2. Results

### 2.1. Primary Cell Cultures

The primary cultures of ATMSCs, WJMSCs, SHEDs, and chondrocytes were successfully obtained and characterized. During expansion in the 2D culture, all MSCs exhibited similar spindle-shaped morphology, sometimes referred to as «fibroblast-like» and typical of cells of mesenchymal origin in an adherent state (Figure 1). MSCs from different tissues were examined for surface antigen expression using flow cytometry. The range of antigens was chosen in accordance with the International Society for Cellular Therapy recommendations [9]. Cells from every source were found to be positive for CD73, CD90, CD44, and CD49b, and negative for hematopoietic markers, thus meeting the MSC immunophenotype criteria (Figure 2). Next, the multilineage differentiation capacity was tested. Osteogenic differentiation was evidenced in all MSC cultures by positive staining of the mineral matrix with alizarin red, while after adipogenic induction, oil red staining revealed the presence of lipid vacuoles (Figure 3). The performed characterization demonstrated that multipotent MSCs were indeed predominant in the ATMSC, WJMSC, and SHED cultures used in this study.

### 2.2. Spheroid Chondrogenic Differentiation

The in vitro chondrogenic potential of ATMSCs, WJMSCs, and SHEDs was compared to that of primary chondrocytes. Chondrogenic differentiation was performed in the 3D spheroid culture to allow cartilaginous ECM deposition and cell–cell interactions. The cells of all types formed spheroids in the ULA plates.

### 2.3. ECM Deposition in MSC Spheroids

Our histological evaluation revealed variations in spheroid morphology across different cell types (Figure 4). The chondrocyte spheroids exhibited two distinct regions, including a peripheral zone plugged with cells and a loose central zone. Spheroids formed of ATMSCs displayed several identifiable zones. A surface zone of densely packed morphologically heterogeneous cells covered a less dense zone characterized by the accumulation of extracellular matrix (ECM). Further inside, a zone of loosely arranged fibroblast-like cells was observed. The very central region of the spheroids was occupied by densely packed polymorphic cells. An examination of the spheroids made up of the WJMSCs revealed a lack of clear zonation, which distinguished them from the chondrocyte and ATMSC-formed spheroids described above. The SHED spheroids were smaller in size than the others, and exhibited a relatively dense overall structure and no distinct zone separation, with the central core being less dense than the periphery.

Histochemical analysis demonstrated that ECM accumulated in all types of spheroids (Figure 4), but its amount and composition differed among them. Collagen in ECM was detected by Masson staining; Alcian blue staining was applied to detect glycosaminoglycan (GAG) synthesis as an indicator of cells undergoing chondrogenic differentiation. On day 14, the highest accumulation of ECM was observed in chondrocyte spheroids. Among MSC spheroids, ECM accretion was greatest in the ATMSC spheroids.

The results obtained on day 21 of culturing spheroids in chondrogenic differentiation medium are of particular interest. ECM accumulation was found to be especially pronounced in chondrocyte spheroids. In these spheroids, specific areas were observed which had a higher ratio of ECM to cells. Chondrocytes in these areas were not tightly surrounded by matrix, which is characteristic of juvenile chondrocytes. Considerable ECM accumulation also occurred in ATMSC spheroids. On the contrary, no significant ECM accumulation occurred in spheroids formed of SHEDs or WJMSCs.

For all spheroid types, the collagen production in ECM was higher on day 21. Masson staining for collagen showed the strongest positive reaction in ATMSC-derived spheroids. Collagen fibers were notably visible at the periphery of chondrocyte and ATMSC spheroids, as well as in the central region of WJMSC spheroids. In SHED spheroids, the collagen staining was of low intensity.

The GAG content in all spheroids increased throughout the culture time, and its highest level was observed in chondrocyte spheroids on day 21. Among MSC spheroids, the ECM in ATMSC and WJMSC spheroids was stained more intensely than in SHED spheroids.

Thus, when cultured in chondrogenic differentiation medium, all spheroids showed GAG accumulation increasing from day 14 to day 21. The radial heterogeneity of collagen distribution can be associated with the inefficient diffusion of nutrients and differentiation inducers inside the spheroids.

### 2.4. Collagen II Synthesis

To more precisely investigate whether the ECM composition in the differentiated spheroids was indeed close to that of the hyaline cartilage, immunohistochemical staining for collagen II was performed on day 21 of differentiation (Figure 5). The results revealed significant accumulation of this cartilage-specific protein in ECM produced by chondrocytes, as well as ATMSCs. The strongest immune reaction appeared to be localized in the same regions of spheroid cross-sections where the most prominent collagen and GAG deposition was detected. Pronounced staining was also observed in the case of WJMSCs, but contrary to the chondrocyte and ATMSC spheroids, the chromogen was mainly bound to the cell cytoplasm rather than distributed in the extracellular space. This pattern might correspond to an earlier stage of chondrogenesis when synthesized collagen molecules have not been secreted yet. Among all the studied cell types, the SHED-based spheroids showed the lowest intensity of staining, which indicated an almost complete absence of collagen II, probably except minor traces visible in the cell cytoplasm.

### 2.5. Biochemical Analysis of GAG/DNA Content

Chondrogenic differentiation was also quantitatively assessed through biochemical assays for DNA and GAG content. 

The DNA content, reflecting the cell proliferation rate, was similar between spheroids made of ATMSCs, SHEDs, and chondrocytes, but it was higher in the WJMSC spheroids (by 2.1–2.3 times and 1.8–2.4 times on day 14 and day 21 of culture with TGF-β1, respectively (*p* < 0.05)) (Figure 6). Note, that the amount of DNA and the diameter of the spheroids do not always conform (Figure 4). This is because the size of the spheroid depends not only on the number of cells, but also on the amount of ECM accumulated during chondrogenic differentiation.

SHEDs produced fewer GAGs per spheroid than chondrocytes on all days of observation (*p* < 0.05). In contrast, in 21-day-old ATMSC spheroids and WJMSC spheroids, the GAG amount was 273.3 ± 56.1 and 181.0 ± 39.6 µg GAGs/spheroid, respectively, and did not significantly differ from that in chondrocyte spheroids (246.6 ± 47.3 µg GAGs/spheroid). 

As seen in Figure 6, the histogram of GAG/DNA content shows similar trends to that of GAG production per spheroid. On day 21, the determination of GAG/DNA content in ATMSC spheroids (0.60 ± 0.17 GAG, µg/DNA, ng) and chondrocyte spheroids (0.63 ± 0.11 GAG, µg/DNA, ng) gave essentially identical values. The lower GAG/DNA content (0.19 ± 0.03 GAG, µg/DNA, ng) observed in WJMSC spheroids indicates increased proliferative activity and reduced differentiation activity.

### 2.6. Changes in Cartilage-Specific Gene Expression

The expression of the genes activated during the cartilage build-up was assessed by quantitative RT-PCR. The study design is shown in Figure 7. We examined the expression of the following genes: Collagen I alpha 1, Collagen II alpha 1, Aggrecan, and SOX-9. The levels of TGF-β1-induced gene transcription in the 3D spheroid cultures of differentiating ATMSCs, WJMSCs, SHEDs, and primary chondrocytes were measured before applying TGF-β1 (day 0), on day 14 and day 21 of chondrogenic differentiation, compared to the levels of the same genes’ transcription in the corresponding two-dimensional cultures. The data obtained are presented in Figure 8.

Aggrecan and Sox-9 expression were upregulated in all spheroid cultures over the entire period of the experiment. Although the kinetics of Collagen II alpha 1 expression in response to TGF-β1 varied between the spheroids investigated, their expression was upregulated in all spheroids from day 0 to day 21, with the maximum expression on day 21. The observed increase occurred gradually in the case of WJMSC-, SHED-, and chondrocyte-based spheroids, while for ATMSC spheroids, it rose abruptly from day 14 to day 21.

Collagen I alpha 1 expression in ATMSC spheroids was downregulated from day 0 to day 14. In WJMSC spheroids, it was not significantly changed on day 0, and was slightly downregulated on day 14. On day 21, the expression of Collagen I alpha 1 in both ATMSC and WJMSC spheroids was only slightly upregulated. At the same time, its expression was clearly upregulated in SHED and chondrocyte spheroids.

## 3. Discussion

The use of MSCs for the tissue engineering of hyaline cartilage has already been successfully implemented in clinical practice and proven to be safe and generally effective [15,16,17]. However, the optimal cell type, culture conditions, and application schedule remain to be elucidated. The generation of hyaline cartilage by MSCs requires combining the following key elements: creating conditions for the chondrogenic differentiation of MSCs, prompting the cells to increase the synthesis of hyaline cartilage ECM proteins, and activating cell proliferation. Various techniques were tested to achieve this goal, including the addition of protein factors (TGF-β superfamily members, bone morphogenetic proteins, insulin-like growth factor-I, fibroblast growth factor, proteoglycans), the formation of 3D structures by using scaffold-free or scaffold-based technologies, and the application of various physical agents (mechanical impact, hypoxia, electromagnetic radiation (photobiomodulation)) [18,19,20]. The selection of the optimal type of MSC with the highest chondrogenic capacity was the aim of the present study.

This paper describes the assessment of the practicality of three types of MSCs as substitutes for chondroblasts in the development of cell technologies for joint cartilage repair. We concentrated on the comparison of the ability of MSCs from varying sources to form chondrosphere-like spheroids and to produce the components of hyaline cartilage in 3D spheroid culture. Our results show that ATMSCs and, to a lesser extent, WJMSCs possess this ability, while SHEDs are probably less suitable. The presented data are generally in line with those reported by other researchers, though not with all of them. This research aimed to determine the ‘ideal’ source of MSCs for cartilage tissue engineering, which is ongoing, and various types of MSCs were shown to assist in articular cartilage repair. Mélou et al., using adult dental pulp MSCs, demonstrated that a spheroid model ensures higher expression of chondrogenesis markers compared to monolayer or pellet culture [21]. By reviewing the reported clinical trials, Peng et al. found that osteoarthritis cell-based therapy translational research is mostly limited to bone marrow-derived MSCs (BMMSCs), ATMSCs, and WJMSCs [22]. BMMSCs are known to have high chondrogenic potential and are efficient in the stimulation of articular cartilage regeneration [23], but the harvesting of MSCs from bone marrow is an invasive procedure prone to complications. In addition, bone marrow aspirate contains only approximately 1 MSC per 105 cells, and a prolonged expansion step is needed to reach the quantity of cell material required in clinical applications [24]. Moreover, the BMMSC number and its differentiation potential decreases with age [25,26]. AT is a promising source of MSCs owing to the simplicity of cell isolation, sufficient cell yield, and limited traumatization during the AT collection procedure. ATMSCs were shown to exert positive therapeutic effects in clinical trials when implanted into a knee joint affected by OA [27]. WJMSCs and dental pulp MSCs are also attractive for regenerative medicine applications, because the process of collecting these cells is noninvasive and their use is not ethically restricted, as the umbilical cords after birth and extracted teeth are considered medical waste. WJMSCs were shown to have the same chondrogenic capacity as BMMSCs [28]. In another study, their population-doubling rate was shown to be twice as high as that of BMMSCs and 1.7 times higher compared to that of ATMSCs [29]. WJMSCs not only exert high proliferative activity, but also demonstrate low immunogenicity, possess antifibrotic activity, and maintain their phenotype during long-term culture [30]. Dental pulp MSCs have high proliferative activity and exert anti-inflammatory effects [31]. The chondrogenic potential of dental pulp MSCs was demonstrated in vivo when cells seeded in a collagen-containing matrix were implanted into the damaged knee joints of minipigs, and in vitro, where cells were cultured on a porous chitosan-xanthan matrix scaffold [32,33]. It should be noted that though a vast variety of MSCs have been studied, only sporadically have direct comparisons of different MSCs been made. This, of course, greatly complicates the choice of the optimal source of MSCs for cell-based therapy.

A number of important features of spheroid formation from different types of MSCs can be evaluated based on the results of the present study. One of them is the different availability of nutrients to cells within the core and the shell of a spheroid. In our study, all MSC spheroids, except WJMSCs, were characterized by radial heterogeneity, which can be attributed to the gradient of diffusion of nutrients, gases, and stimulating factors from the medium. Really, the spatially uniform distribution of specific ECM proteins can be achieved by reducing the spheroid size [34]. Three-dimensional culture in perfusion bioreactors is an alternative way to achieve the homogeneity of the cell layer structure due to the uniform supply of nutrients and gases to the cells, as well as the more efficient removal of metabolic waste products [35,36]. Homogeneous ECM production is promoted by culturing cells on a matrix designed to replicate in vitro the native tissue architecture [37], which has been demonstrated, for example, by the formation of cartilage-like structures on the microparticles of decellularized articular cartilage [38]. However, despite the lack of structural homogeneity, in vitro-grown spheroids composed of autologous chondrocytes can be successfully used in the treatment of knee joint injuries [39]. Knee joint repair in rats was performed by injecting spheroids of dedifferentiated adipocytes into the osteochondral defect [40]. Nevertheless, homogeneous ECM distribution is likely to improve the quality of tissue equivalent materials for implantation.

Interestingly, the results of the present study are in line with those obtained by Kim et al. for porcine ATMSCs and WJMSCs. Porcine ATMSCs exhibited a higher proliferation rate and more efficient differentiation into adipogenic and chondrogenic lineages than porcine WJMSCs, while WJMSCs showed a superior ability to differentiate into osteogenic lineages [41]. The results obtained on animal MSCs, though, are not necessarily similar to those for human MSCs. Thus, Merlo et al. revealed the difference in properties between equine and human WJMSCs [42]. There may be differences between MSCs from infants and adults. Indeed, higher chondrogenic potential of infant ATMSCs compared to adult cells was reported [43].

The aging of adult MSC populations is a well-known phenomenon [44]. Over the cells’ lifetime, multiple intrinsic and extrinsic factors contribute to the aging process, including signaling pathways, cytokines, chemokines, growth factors, hormones, vitamins, drugs, chemicals, and environmental factors [45]. As a result, this leads to diminished self-renewal and differentiation capacity of older MSCs, reducing their therapeutic potential. This problem does not exist in the case of WJMSCs, young MSCs, which confers upon these cells a significant clinical advantage over other senescent MSCs. Given the particularly pronounced immunomodulatory properties of WJMSCs, these cells may be suggested to be perfectly suitable for allogenic transplantations, thus enabling cell therapy for elderly patients whose autologous cultures cannot be obtained [46].

In this study, we investigated the chondrogenic potential of different MSC types in three-dimensional culture under the influence of TGF-β1. We also attempted to compare these results to those obtained using growth medium. However, in the absence of differentiation induction, the proliferation of cells inside spheroids did not slow down, leading to spheroid overgrowth, the formation of necrotic cores, and, finally, disaggregation on days 10–14.

The expression of the chondrocyte-specific markers was assessed by RT-qPCR.

The level of transcription of Aggrecan, a cartilage proteoglycan, increased already on day 0 of cultivation with an inducer in each of the three-dimensional cultures, and also increased further, reaching a maximum on day 21. The expression of Sox-9, a transcription factor expressed at the early stage of chondrogenesis, showed a similar trend. The upregulation of both Aggrecan and Sox-9 expression is generally considered a confirmation of chondrogenic differentiation taking place in the 3D cultures [47].

Collagen II alpha 1 is one of the main components of articular hyaline cartilage ECM [48]. Collagen II transcription level increased in all spheroids studied, and reached a maximum on day 21. The most pronounced increase on day 21 was observed in the chondrocyte spheroids, while the highest increase in the transcription levels on day 0 and day 14 occurred in the ATMSC and WJMSC spheroids.

Collagen I is a major fibrillar component of undifferentiated mesenchymal progenitor cells. After differentiation, chondrocytes cease to produce collagen I, and start to produce typical cartilage components, including collagen II and aggrecan [49]. Our results also show increased expression of COL1A1 in 3D culture compared to 2D culture. A decrease in the level of COL1A1 transcription on day 14 compared to day 0 in ATMSCs and WJMSCs indicates a stronger effect of TGF-β1 and, accordingly, more pronounced further differentiation along the articular cartilage-formation pathway.

As shown in the work by Kim et al. [50], culturing cells in 3D conditions is by itself sufficient for the enhancement of the expression of ACAN, SOX9, and COL2AI. These genes are also markers of the formation of articular cartilage. However, it was reported that further cultivation in 3D conditions in the absence of differentiation inducers results in the osteogenic differentiation of MSCs [50].

The comparison of the transcription level changes of Collagen II and Collagen I shows that along with the more potent Collagen II upregulation in ATMSC and WJMSC on day 0 and day 14, the transcription levels of Collagen I either changed within narrower limits (days 0 and 21 for WJMSC and day 0 for ATMSC) or even decreased (day 14 for ATMSC). In addition, the increase in collagen I transcription levels on day 21 for ATMSC and WJMSC was also less than for chondrocytes.

It was expected that the cells cultured using the 3D spheroid culture methods would express chondrogenic markers such as Collagen II, Aggrecan, and SOX-9, but relatively low levels of fibrous markers. The upregulation of chondrogenic markers along with the downregulation or slight upregulation of hypertrophy and fibrosis-related genes may indicate hyaline cartilage-like instead of hypertrophic cartilage-like spheroid differentiation.

Thus, according to the data obtained with the set of chondrogenesis marker genes, including Collagen I alpha 1, Collagen II alpha 1, Aggrecan, and SOX-9, it is confirmed that ATMSC spheroids are most suitable for the regenerative therapy of articular cartilage using tissue engineering methods. In addition, it was found that WJMSC can also be used for the restoration of articular cartilage.

In conclusion, MSCs, in addition to being prone to chondrogenic differentiation, have other properties of practical significance in the field of cartilage restoration, including immunosuppressive and anti-inflammatory activity. Our results suggest that among the MSC types investigated, ATMSCs, when cultured as spheroids, have the highest chondrogenic potential. WJMSCs also appear to be a promising cell source for cartilage tissue engineering. It was also found that the transfer of cells from 2D to 3D spheroid culture by itself, even without adding a chondrogenic inducer, triggers chondrogenesis-related marker gene expression.

## 4. Materials and Methods

### 4.1. Primary Cell Cultures

WJMSCs were isolated from umbilical cords collected after normal deliveries (n = 3; W 28, W 33, W 31). Fragments of the umbilical cords were wiped with 70% ethanol tissues and washed twice in a 50 mL tube with HBSS (Gibco, Carlsbad, CA, USA) supplemented with antibiotic/antimycotic solution (Gibco, Carlsbad, CA, USA). The tissue was then placed in a Petri dish containing 4 mL 0.1% collagenase I solution (Gibco, Carlsbad, CA, USA), minced mechanically to a particle size of ~1 mm^3^, and incubated for 45 min at 37 °C. Cells were precipitated by centrifugation and resuspended in the growth medium (DMEM/F12 supplemented with 10% fetal bovine serum and 100 units/mL penicillin/streptomycin (all from Gibco, Carlsbad, CA, USA).

Adipose tissue samples weighing 3–5 g were obtained from healthy donors during living related liver transplantation under general anesthesia (n = 3; M 19, M 24, W 26). The samples were rinsed twice with HBSS (Gibco, Carlsbad, CA, USA) supplemented with antibiotic/antimycotic solution (Gibco, Carlsbad, CA, USA), placed in 0.1% collagenase I solution, and incubated for 20 min at 37 °C. Cells were harvested by centrifugation and resuspended in the growth medium.

Human deciduous teeth were collected from healthy children after normal exfoliation (n = 3; M 5, M 5, M 8). Samples were stored in HBSS containing an antibiotic/antimycotic (Gibco, Carlsbad, CA, USA) until delivery to the laboratory within 24 h. Pulp tissue was mechanically extracted from the crown, disintegrated, and digested in 0.1% collagenase type I solution in HBSS (Gibco, Carlsbad, CA, USA) for 60 min at 37 °C. Cells were harvested by centrifugation and resuspended in the growth medium.

Hyaline articular cartilage was sampled from the intact articular surface of the femoral condyle taken intraoperatively during knee joint endoprosthesis surgery (n = 3; M 24, M 27, W 38). The samples were washed twice in HBSS (Gibco, Carlsbad, CA, USA) with antibiotic/antimycotic (Gibco, Carlsbad, CA, USA), mechanically disintegrated into pieces less than ~1 mm^3^), placed in 0.1% collagenase I solution, and incubated under standard conditions for 60 min. After centrifugation, the cells were transferred alongside with predigested tissue to culture flasks containing growth medium.

The cells were expanded in 75 cm^2^ flasks under standard conditions to 80% confluency. For passaging, the cells were detached by incubating with 0.25% trypsin-EDTA solution (Paneko, Moscow, Russia) for 5–10 min at 37 °C, washed twice with PBS, and subcultured at a 1:3 ratio in the growth medium.

### 4.2. Flow Cytometry

Primary cell cultures were subjected to flow cytometry analysis to determine the surface expression of CD markers typical of MSCs.

After trypsinization, 50 μL aliquots of a single-cell suspension (2 × 10^5^ cells/mL in PBS with 2% fetal bovine serum) were stained with 5 µL of fluorescently labeled monoclonal antibodies against CD73, CD90, CD44, CD49b, or a cocktail of antibodies to hematopoietic markers (CD45, CD34, CD14, CD11b, CD79a, CD19, HLA-DR), and conjugated with phycoerythrin (PE) or fluorescein isothyocyanate (FITC) (BD Biosciences, San Diego, CA, USA). The cells were washed twice with PBS and fixed with BD Cytofix™ solution (BD Biosciences, San Diego, CA, USA). The samples were analyzed using a FACS Aria III flow cytometer (BD Biosciences, San Diego, CA, USA).

### 4.3. Osteogenic Differentiation

Cells were seeded in 6-well plates (Corning, Corning, NY, USA) (100,000 cells/well) and cultured in growth medium until they reached 80–90% confluency. Next, the medium was changed to osteogenic differentiation medium (DMEM containing 2 mM L-glutamine, 100 U/mL penicillin, 100 U/mL streptomycin (all from Gibco, Carlsbad, CA, USA), 0.2 mM ascorbic acid, 10 mM glycerophosphate, and 0.1 μM dexamethasone (all from Sigma-Aldrich, St. Louis, MO, USA)). The osteogenic medium was refreshed every three days. After 14 days of differentiation, cells were fixed with 4% paraformaldehyde (PFA) and stained with 2% alizarin red (Sigma-Aldrich, St. Louis, MO, USA) to detect mineral (calcium) deposition.

### 4.4. Adipogenic Differentiation

Cells were seeded in 6-well plates (Corning, Corning, NY, USA) (100,000 cells/well) and cultured in growth medium until they reached 80–90% confluency. Next, the medium was changed to adipogenic medium (DMEM supplemented with 10% horse serum, 0.5 mM isobutylmethylxanthine, and 60 mM indomethacin (all from Sigma-Aldrich, St. Louis, MO, USA)). The adipogenic medium was refreshed every 3 days. After 14 days of differentiation, cells were fixed with 4% PFA and stained with oil red O (Sigma-Aldrich, St. Louis, MO, USA) to detect intracellular lipid droplets.

### 4.5. Chondrogenic Differentiation

Cell spheroids were made by aggregating the cells in 96-well ultra-low attachment (ULA) plates (Corning, Corning, NY, USA). For this purpose, 200 μL aliquots of cell suspension in growth medium (5 × 10^5^ cells/mL) were placed in the wells producing spheroids consisting of 100,000 cells/spheroid. Cell cultures in passages 4–6 were used for the experiments.

After 2 days of culture in standard conditions, the growth medium was replaced with the chondrogenic induction medium (high-glucose DMEM (Gibco, Carlsbad, CA, USA), 10% ITS + Premix (Corning, Corning, NY, USA), 1% sodium pyruvate (Sigma-Aldrich, St. Louis, MO, USA), 0.25% ascorbate-2-phosphate (Sigma-Aldrich, St. Louis, MO, USA), 0.1 μM dexamethasone (Sigma-Aldrich, St. Louis, MO, USA), 1% fetal bovine serum (Gibco, Carlsbad, CA, USA), 1% penicillin-streptomycin-glutamine (Gibco, Carlsbad, CA, USA) and 10 ng/mL TGF-β1 (PeproTech, Rocky Hill, CT, USA)). The spheroids were cultured in chondrogenic medium for 21 days with daily medium change.

### 4.6. Sample Lysis, RNA Extraction, Reverse Transcription, and qPCR Analysis

Direct cell lysis was performed using the QIAzol reagent (Qiagen, Hilden, Germany) according to the manufacturer’s instructions. Spheroid samples were first flash-frozen in liquid nitrogen and homogenized with a mini hand-held homogenizer, immediately placed on ice for thawing, and then lysed by adding the lysis agent. RNA extraction from lysates was performed using the RNeasy Lipid tissue mini kit from Qiagen according to the manufacturer’s instructions. Purified total RNA was eluted in RNase-free water and stored at −20 °C for downstream analysis.

The RT2 qPCR Primer Assays (Qiagen, Hilden, Germany) used in this work are listed in Table 1 with their Qiagen catalogue numbers. 

A MxPro 3005P (Stratagene, La Jolla, CA, USA) real-time system and MxPro 3000 software were used for real-time transcription analysis of collagen type I alpha 1, collagen type II alpha 1, and aggrecan encoding genes. The QuantiTect Reverse Transcription Kit and the QuantiTect SYBR Green PCR Kit were used. All samples were analyzed in duplicate, and the mean value was used for further analysis. The mean cycle threshold (Ct) values were normalized against the internal control genes glyceraldehyde-3-phosphate dehydrogenase (GAPDH) to calculate gene expression according to the 2^(−ΔΔCt)^ method.

### 4.7. DNA Quantification

DNA was isolated using the DNeasy Blood & Tissue Kit (QIAGEN, Hilden, Germany) according to the manufacturer’s instructions. Spheroids were digested with Lysis Buffer (Buffer AL, QIAGEN, Hilden, Germany) and proteinase K for 16 h at +57 °C. For maximum DNA yield, elution was performed in three successive steps using 200 µL Elution Buffer (Buffer AE, QIAGEN, Hilden, Germany) each time.

Quantitative measurements of the total DNA content were then carried out using the Quant-IT PicoGreen dsDNA Assay Kit (Invitrogen, Waltham, MA, USA) according to the manufacturer’s specifications and a Spark 10 M microplate reader (TecanTrading AG, Switzerland). A standard curve was obtained by measuring the fluorescence signal of bacteriophage λ DNA solutions with concentrations from 0.0 to 1000.0 ng/mL.

### 4.8. Biochemical Quantification of GAG Content

Spheroids were digested overnight using papain solution (125 µg/mL papain in PBS at pH 6.5 with 5 mM L-cysteine hydrochloride, and 5 mM EDTA (all reagents Sigma-Aldrich, St. Louis, MO, USA)). The digested samples were then centrifuged at 10,000× *g* for 10 min, and 1,9-dimethylmethylene blue solution (200 µL) was added to each standard concentration of chondroitin sulphate and the sample (20 µL) in a clear flat bottomed 96-well plate (all reagents Sigma-Aldrich, St. Louis, MO, USA). The plate was agitated for 2 min before the absorbance was measured with a Tecan infinite M200 Pro plate reader (Tecan Trading AG, Männedorf, Switzerland) at a wavelength of 525 nm.

### 4.9. Histological and Immunohistochemical Staining

Samples were preserved in 10% formalin solution, dehydrated, paraffinized, sectioned at 4–5 μm, deparaffinized in xylene, rehydrated, and stained with hematoxylin and eosin (H&E), alcian blue, and with Masson’s staining.

Collagen II in the samples was visualized using a Rabbit Specific HRP/DAB(ABC) Detection IHC Kit (Abcam, Cambridge, UK) and antibodies for collagen II (Abcam, Cambridge, UK). All the reagents listed below were from Abcam, Boston, MA, USA. Paraffin sections were deparaffinized and subjected to Hydrogen Peroxidase Block treatment for 10 min in order to neutralize endogenous peroxidase. The samples were washed twice with Tris-Buffered Saline (TBS). In order to unmask an antigen, they were subjected to a preliminary hot-temperature treatment in Citrate Buffer pH 6.0 at 95 °C in a water bath for 20 min. The slices were cooled for 20 min at room temperature and washed three times with TBS. In order to prevent non-specific binding, samples were incubated with a Protein Block for 10 min. After washing twice, primary antibodies were applied to the section (Rabbit polyclonal antibodies for collagen 2 used at a 1:100 dilution) and the samples were incubated overnight at 4 °C. Then, they were washed four times, and secondary antibodies (Biotinulated Goat Anti-Polyvalent) were applied to the section for 10 min at room temperature. After this, the samples were washed four times and incubated with Streptavidin Peroxidase for 10 min with subsequent washing four times, followed by application of the dye substrate (DAB Chromogen) for 2–5 min. The samples were washed four times with TBS, rinsed in distilled water, contrasted with hematoxylin, dehydrated, and placed in balsam. The staining results were studied with a Nikon Eclipse microscope (Nikon Corp, Tokyo, Japan).

### 4.10. Statistical Analysis

Data were analyzed with SPSS26.0 statistical software package. The results were presented as mean ± SD. The distribution of the variables was tested with the Shapiro–Wilk procedure. The results were compared using one-way analysis of variance (ANOVA) and Tukey’s honestly significant difference post hoc test, where *p* < 0.05 was considered statistically significant.

## Figures and Tables

**Figure 1 ijms-25-05695-f001:**
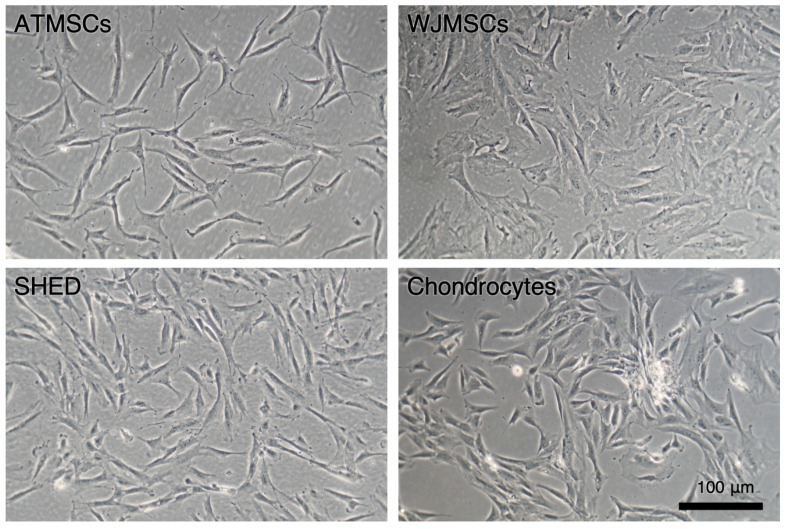
Primary adherent cultures of human ATMSCs, WJMSCs, SHEDs, and chondrocytes. Phase-contrast microscopy.

**Figure 2 ijms-25-05695-f002:**
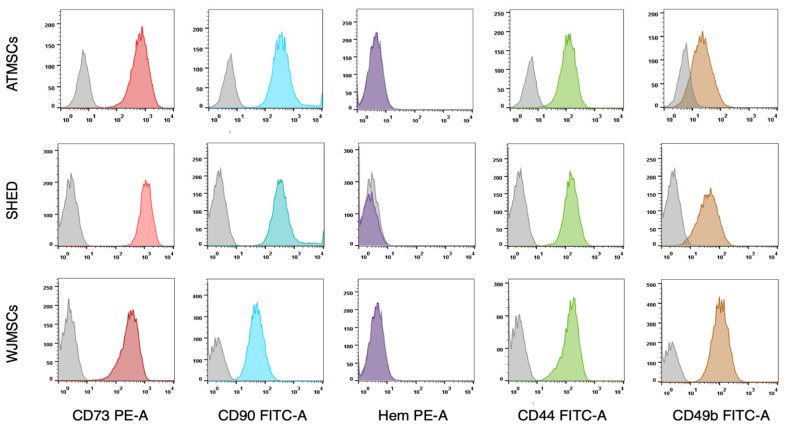
Expression of surface molecular markers by MSCs isolated from different tissues. Cell suspensions of ATMSCs, SHED, and WJMSCs were treated with fluorochrome-conjugated monoclonal antibodies and analyzed by flow cytometry. Grey curves—isotype-negative controls. Colored curves correspond to specific molecular markers. Hem—cocktail of antibodies for hematopoietic markers (CD45, CD34, CD14, CD11b, CD79a, CD19, and HLA-DR).

**Figure 3 ijms-25-05695-f003:**
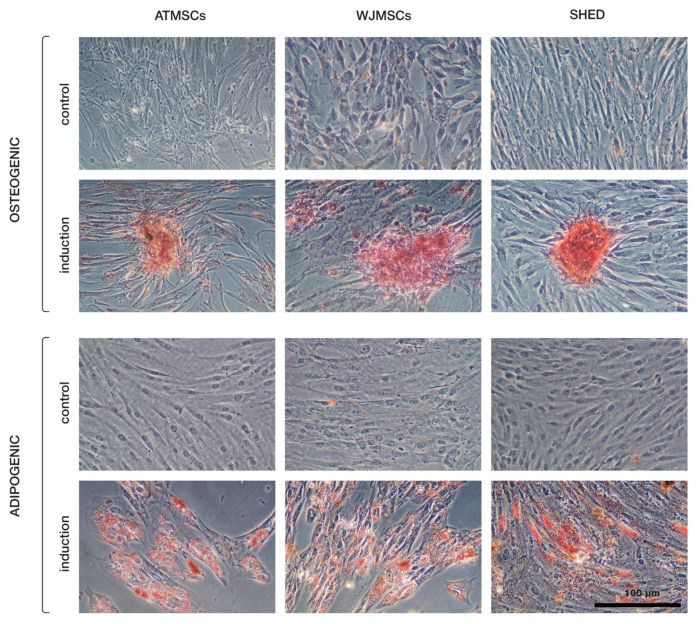
Multilineage differentiation of ATMSCs, WJMSCs, and SHEDs. Osteogenic differentiation (**upper rows**) was assessed using alizarin red S staining, and adipogenic differentiation (**bottom rows**) using oil red O.

**Figure 4 ijms-25-05695-f004:**
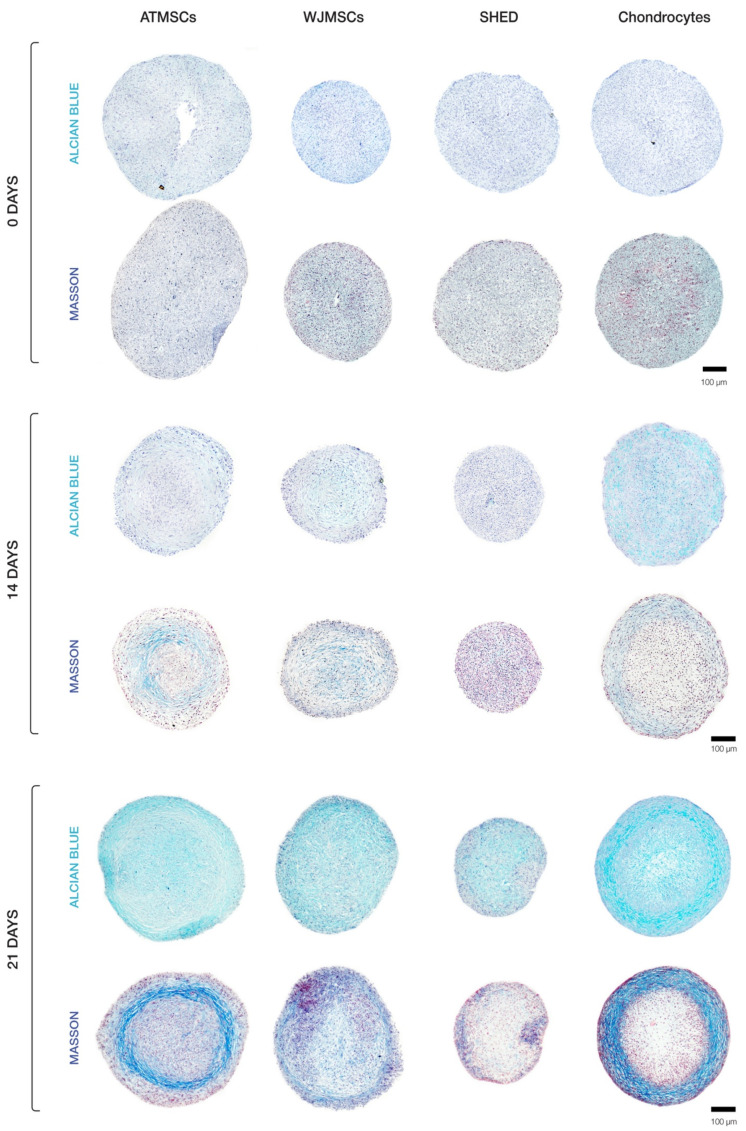
Histological analysis of 3D spheroids undergoing chondrogenic differentiation.

**Figure 5 ijms-25-05695-f005:**
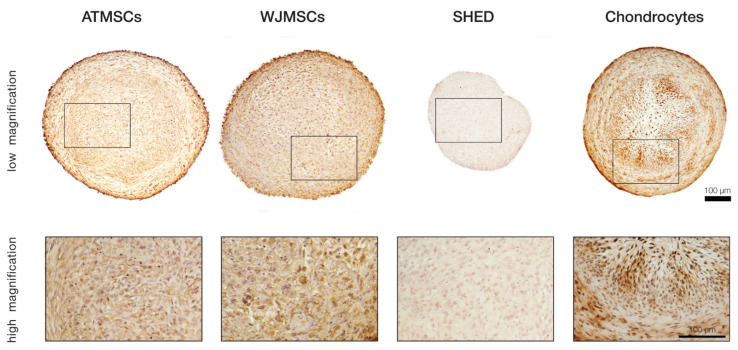
Immunohistochemical analysis of collagen type II in differentiated 3D spheroids.

**Figure 6 ijms-25-05695-f006:**
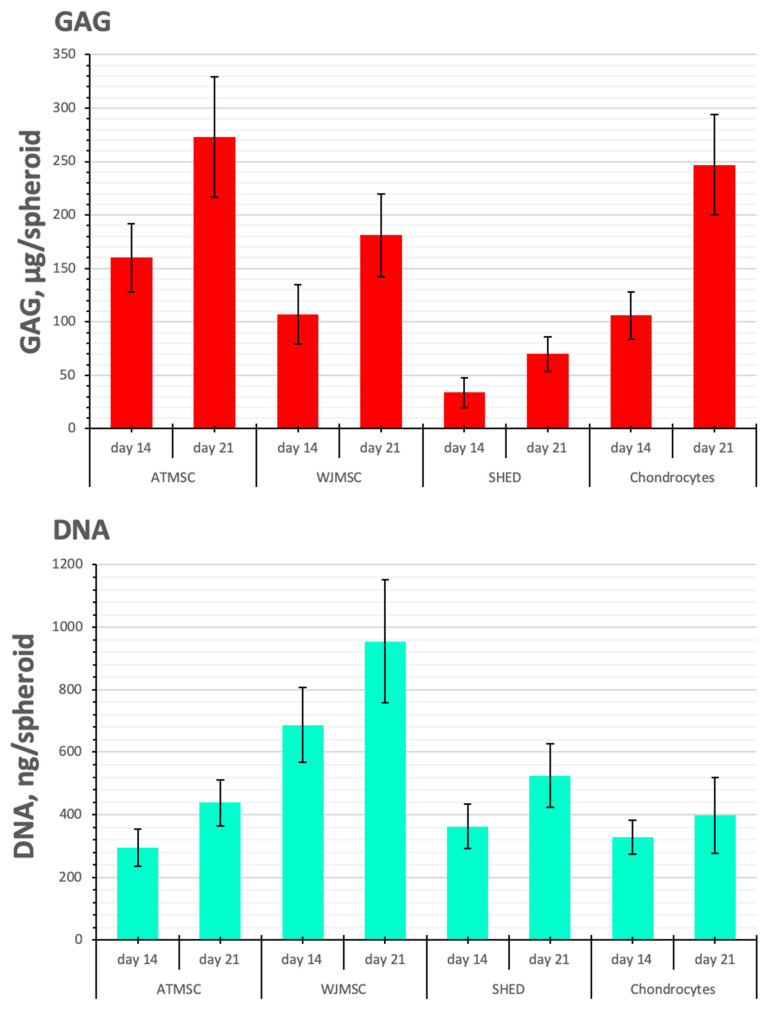
GAG/DNA content in ATMSC, WJMSC, SHED, and chondrocyte spheroids during culture with TGF-β1.

**Figure 7 ijms-25-05695-f007:**
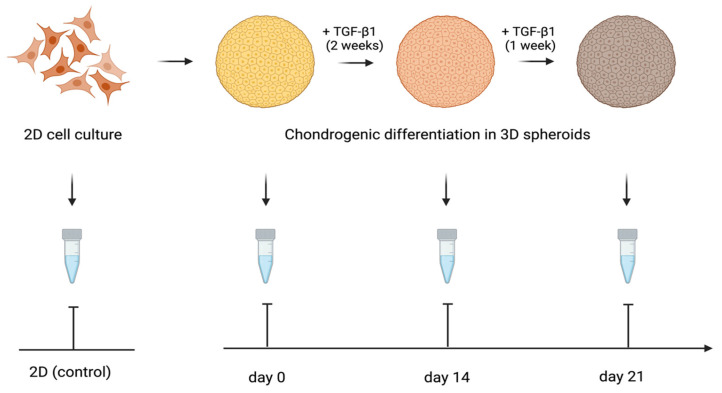
Design of the study of cartilage-specific gene expression in 3D spheroids undergoing chondrogenic differentiation.

**Figure 8 ijms-25-05695-f008:**
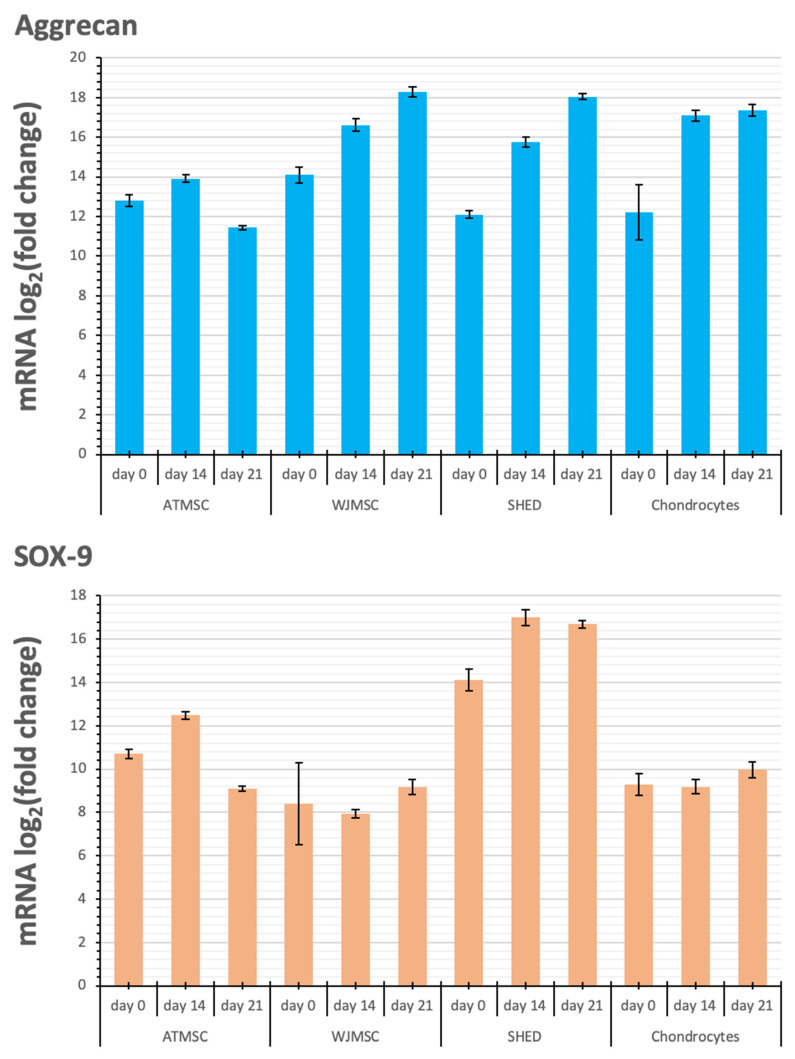
Changes in transcription levels of the genes studied (log2 of the fold change) within 3D spheroids of different cells before applying TGF-β1 (day 0) on day 14 and day 21 of chondrogenic differentiation compared with 2D cultures in growth medium.

**Table 1 ijms-25-05695-t001:** List of gene primers and their Qiagen catalogue numbers.

Gene Symbol and Name	Qiagen Catalogue Number
GAPDH, glyceraldehyde-3-phosphate dehydrogenase	330001 PPH00150F
Col1A1, Collagen, Type I, alpha 1	330001 PPH01299F
Col2A1, Collagen, Type II, alpha 1	330001 PPH02134F
ACAN, Aggrecan	330001 PPH06097E
SOX-9, SRY (sex-determining region Y)-box 9	330001 PPH02125A

## Data Availability

The original contributions presented in the study are included in the article, further inquiries can be directed to the corresponding author.

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
