# Peer review of "Cartilage-Specific Gene Expression and Extracellular Matrix Deposition in the Course of Mesenchymal Stromal Cell Chondrogenic Differentiation in 3D Spheroid Culture"

_ijms, 2024, doi:10.3390/ijms25115695_

Round 1
Reviewer 1 Report (Previous Reviewer 1)
Comments and Suggestions for Authors
The manuscript entitled “Cartilage-Specific Gene Expression and ECM Deposition in the Course of MSC Chondrogenic Differentiation in 3D Spheroid Culture” described the chondrogenesis capability of mesenchymal stromal cells derived from different sources in 3D spheroid culture ex vivo. The topic is not novel, and the revised manuscript is similar as the original version. There are some issues that need to be addressed in the manuscript:
1. The expression of collagen II increased overtime in all four groups, collagen I showed a similar trend between ATMSC and WJMSC groups, but aggrecan and SOX9 was highly expressed in SHED group. To make it more appealing study for articular cartilage repair application, authors can detect specific gene and protein expression of hyaline cartilage in those spheroids, to further demonstrate which MSC source is superior, such as IGF2 and IGL@ that are highly expressed in hyaline cartilage (can refer to https://www.nature.com/articles/jhg200331).
2. Line 129-131: Please provide the cell size measurement if the authors think the cell size has an effect on the spheroid formation. From the brightfield images provided in Figure 1 and 3, the cell size seems to be similar between different tissue donor sites. Please provide the average size of spheroids from different groups at 14 ad 21 days. Any difference of fibroblast-like cell population between groups? Fibroblasts tend to cause more ECM shrinkage in in vitro culture and produce dense collagen I ECM (confirmed by the high content of collagen I in SHED group).
3. Collagen X and MMP13 measurement of the spheroid ECM can help to demonstrate cartilage hypertrophia, especially those centrally located chondrocytes in the spheroids.
4. As the authors presented in Figure 3, osteogenic and adipogenic differentiation capacity of those MSCs was proven in 2D culture. How about the chondrogenic differentiation in 2D culture?
Author Response
Dear Editors and Reviewers,
Thank you for reviewing our manuscript, ijms-2817617, entitled «Cartilage-Specific Gene Expression and ECM Deposition in the Course of MSC Chondrogenic Differentiation in 3D Spheroid Culture» and valuable comments and suggestions.
Please, find our point-by-point responses to the reviewers’ comments below.
Remark 1
The expression of collagen II increased overtime in all four groups, collagen I showed a similar trend between ATMSC and WJMSC groups, but aggrecan and SOX9 was highly expressed in SHED group. To make it more appealing study for articular cartilage repair application, authors can detect specific gene and protein expression of hyaline cartilage in those spheroids, to further demonstrate which MSC source is superior, such as IGF2 and IGL2 that are highly expressed in hyaline cartilage (can refer to https://www.nature.com/articles/jhg200331).
Response 1
Thank you for your valuable advice. Indeed, IGF2 play important role in maintaining MSC stemness and forming MSC tissue niche. However, according to recent studies, this genes functions include regulation of NANOG, OCT4 and SOX2 gene expression, as well as activation of MSC ability to differentiate in various directions including adipose and bone tissue (https://www.isct-cytotherapy.org/article/S1465-3249(18)30702-3/abstract). For this reason, the expression level of these molecule is rarely used as a marker of chondrogenic differentiation. However, we will definitely pay attention to it in our ongoing work which among others focuses on MSC transcriptome profile under spheroid culture conditions.
Remark 2
Line 129-131: Please provide the cell size measurement if the authors think the cell size has an effect on the spheroid formation. From the brightfield images provided in Figure 1 and 3, the cell size seems to be similar between different tissue donor sites. Please provide the average size of spheroids from different groups at 14 ad 21 days. Any difference of fibroblast-like cell population between groups? Fibroblasts tend to cause more ECM shrinkage in in vitro culture and produce dense collagen I ECM (confirmed by the high content of collagen I in SHED group).
Response 2.
Unfortunately, the diameter in the histological sections does not exactly correspond to the diameter of the spheroid, because it is technically impossible to ensure that the section passes exactly through the middle of the spheroid. For this reason, we cannot provide the required quantitative information. And, accordingly, in the revised manuscript, we deleted our inappropriate speculation on the effect of cell size on spheroid formation.
Remark 3.
Collagen X and MMP13 measurement of the spheroid ECM can help to demonstrate cartilage hypertrophia, especially those centrally located chondrocytes in the spheroids.
Response 3.
Chondroblast hypertrophy is a sign of osteochondral ossification, which is observed in the growth zone of the tubular bones and which is undesirable in the case of hyaline cartilage. In our work, we used Collagen I as a negative marker of osteochondral ossification. To improve the quality of the evaluation results, its expression was analyzed not separately, but in comparison with the changes in the expression of Collagen II. As in works by others, a decrease in Collagen I expression with a simultaneous increase in Collagen II expression was considered by us as sufficient evidence of the absence of cell hypertrophy.
As you have rightly noticed, on the presented histological images of spheroid sections, there are signs of cell hypertrophy, which is especially pronounced in centrally located chondrocytes. However, this occurred only in spheroids based on chondrocytes (control group). In this regard, we used the above-described widespread method of proving hyaline cartilage formation.
Below please find some examples of works with an approach similar to ours:
https://www.mdpi.com/2306-5354/10/10/1182
https://www.frontiersin.org/articles/10.3389/fbioe.2022.986310
https://doi.org/10.3390/biomedicines11051314
https://www.mdpi.com/2306-5354/11/2/112
https://doi.org/10.3390/ijms21082785
https://doi.org/10.3390/polym15193938
Currently, we have begun to study the transcriptome profile of MSCs in spheroid cultures. It has already been found that the expression level of MMP-13 is reduced in MSC spheroids compared to 2D culture. However, to date, these data were obtained at early time points, and we cannot cite them in the present article
Remark 4
As the authors presented in Figure 3, osteogenic and adipogenic differentiation capacity of those MSCs was proven in 2D culture. How about the chondrogenic differentiation in 2D culture?
Response 4
Since 2D cultures cannot provide optimal conditions for chondrogenic differentiation, they are currently hardly used. Optimal conditions for in vitro chondrogenic differentiation involve three-dimensional aggregation of cells into spheroids, where high cell density and cell-cell interactions play an important role in the mechanism of chondrogenesis. That’s why we believe that 3D spheroid culture provides more physiologically relevant model for studying chondrogenic differentiation, as is also recommended in the methodological articles on this topic (https://www.ncbi.nlm.nih.gov/pmc/articles/PMC3106977/).
Reviewer 2 Report (New Reviewer)
Comments and Suggestions for Authors
We need to know the sex and age of each batch donor. Did all the patients were healthy?
Negative isotope controls for flow cytometry are needed.
What about the concentration of the Ab used?
You should use the dot and not the "," for decimals.
Nothing is said about the number of cells seeded before multilineage differentiation.
Line 449, there is a typo.
There are already papers on the comparison of these MSC from the same sources for chondrogenesis, such as Basok 2019.
Fig 6 should have the same design as Fig 8.
You Saif the cells on the edges look like fibroblasts. Have you got photos of them?
Comments on the Quality of English LanguageModerate revision
Author Response
Dear Editors and Reviewers,
Thank you for reviewing our manuscript, ijms-2817617, entitled «Cartilage-Specific Gene Expression and ECM Deposition in the Course of MSC Chondrogenic Differentiation in 3D Spheroid Culture» and valuable comments and suggestions.
Please, find our point-by-point responses to the reviewers’ comments below.
Responses to Reviewer 2.
Remark 1.
We need to know the sex and age of each batch donor. Did all the patients were healthy?
Response 1.
We have added the required information about the donors [lines 394, 403, 409].
Remark 2.
Negative isotope controls for flow cytometry are needed.
Response 2.
The isotype control is present as a gray curve in all histograms (fig.2). For clarity, we have included "negative" as an explanation of the gray curve in the figure legend.
Remark 3.
What about the concentration of the Ab used?
Response 3.
This information has been added in the Materials and Methods section [lines 428 - 429].
Remark 4.
You should use the dot and not the "," for decimals.
Response 4.
Thank you for pointing this out. The error has been corrected
Remark 5.
Nothing is said about the number of cells seeded before multilineage differentiation.
Response 5.
This information has been added in the text [lines 437, 438].
Remark 6.
Line 449, there is a typo.
Response 6,
Thank you for pointing this out. The error has been corrected.
Remark 7.
There are already papers on the comparison of these MSC from the same sources for chondrogenesis, such as Basok 2019.
Response 7.
Indeed, we have already published a paper presenting the preliminary results demonstrating the enhanced ability of MSCs from adipose tissue to differentiate in the chondrogenic direction. However, the present work has another goal and is focused on obtaining the cartilage-specific gene expression profile of MSCs from different tissues. In addition, in this work we used newly obtained MSC cultures.
Remark 8.
Fig 6 should have the same design as Fig 8.
Response 8.
The design of Figure 6 has been updated and aligned with Figure 8.
Remark 9.
You said the cells on the edges look like fibroblasts. Have you got photos of them?
Response 9.
This statement was made by us only on the basis of the fact that the cells on the surface looked more spread and flattened. We have to agree that it is insufficiently substantiated. This sentence has been deleted from the text.
Round 2
Reviewer 2 Report (New Reviewer)
Comments and Suggestions for Authors
I think the authors have replied my comments. The work would be very stronger and useful if you included the controls for GAGs and collagen II deposition obtained at time 0, especially considering cells without any TGF-B1 show over-expression of cartilage-related genes. Furthermore, controls in normal media, cultivated for the same amount of time, should be included. Ithink it would definitely help the paper to get more and more citations, in the future.
Comments on the Quality of English LanguageMinor
Author Response
Dear Editors and Reviewers,
Thank you for reviewing our manuscript, ijms-2817617, entitled «Cartilage-Specific Gene Expression and ECM Deposition in the Course of MSC Chondrogenic Differentiation in 3D Spheroid Culture» and valuable comments and suggestions.
Please, find our point-by-point responses to the reviewers’ comments below.
Responses to Reviewer 2.
Remark 1.
I think the authors have replied my comments. The work would be very stronger and useful if you included the controls for GAGs and collagen II deposition obtained at time 0, especially considering cells without any TGF-B1 show over-expression of cartilage-related genes.
Response 1.
Following your recommendation, we have added (to Figure 4) the results of histological analysis of samples before starting differentiation (day 0). It should be noted, however, that the data obtained indicate that no specific components of cartilage ECM, including total collagen, accumulated at such an early stage of culture, as indicated by negative Masson staining. In view of this fact, the immunohistochemical staining for collagen II at day 0 was not performed, though the activation of expression of a number of marker genes was observed by RT-PCR.
Remark 2.
Furthermore, controls in normal media, cultivated for the same amount of time, should be included. I think it would definitely help the paper to get more and more citations, in the future.
Response 2.
Indeed, we first tried to culture the spheroids in the growth medium. However, in the absence of differentiation induction, the proliferation of cells inside the spheroids did not slow down, leading to spheroid overgrowth, formation of necrotic cores and, finally, disaggregation on days 10-14. These data were not considered worthy of being included in the manuscript.
Round 3
Reviewer 2 Report (New Reviewer)
Comments and Suggestions for Authors
The reviewers have positively answered my major comments.
I would like to invite the authors to soluble check the correctness of both the concentrations and unit of measures. For instance, in the paragraph of osteogenesis and adipogenesis, "mM" is not written correctly. The reported concentration of indomethacin is 60M. I believe there is a mistake. Furthermore, you need to be consistent. Sometimes you write 0.0001% dexamethasone, other times, 0.1 uM. Please, use the same scientific notation.
I have noticed the use of brackets is often wrong, double check the correctness in the osteogenesis, adipogenesis and chondrogenesis paragraphs.
It would be nice if you include your observations on using growth media (necrosis, pellets lost their structure) in the discussion.
Comments on the Quality of English LanguageCarefully check the syntax and grammar, including the use of brackets.
Author Response
Dear Editors and Reviewers,
Thank you for reviewing our manuscript, ijms-2817617, entitled «Cartilage-Specific Gene Expression and ECM Deposition in the Course of MSC Chondrogenic Differentiation in 3D Spheroid Culture» and valuable comments and suggestions.
Please, find our point-by-point responses to the reviewers’ comments below.
Responses to Reviewer 2.
Remark 1.
I would like to invite the authors to soluble check the correctness of both the concentrations and unit of measures. For instance, in the paragraph of osteogenesis and adipogenesis, "mM" is not written correctly. The reported concentration of indomethacin is 60M. I believe there is a mistake. Furthermore, you need to be consistent. Sometimes you write 0.0001% dexamethasone, other times, 0.1 uM. Please, use the same scientific notation. I have noticed the use of brackets is often wrong, double check the correctness in the osteogenesis, adipogenesis and chondrogenesis paragraphs.
Response 1.
Thank you very much for analyzing our work so carefully. We regret that there were many typos in the final text. These errors have now been corrected, as well as anything else that was found during a final check.
Remark 2.
It would be nice if you include your observations on using growth media (necrosis, pellets lost their structure) in the discussion.
Response 2.
Following your recommendation we’ve included a paragraph in the discussion [lines 336 – 341].
This manuscript is a resubmission of an earlier submission. The following is a list of the peer review reports and author responses from that submission.
Round 1
Reviewer 1 Report
Comments and Suggestions for Authors
The manuscript entitled “Mesenchymal Stromal Cells Isolated from the Umbilical Cord, Adipose Tissue and the Deciduous Tooth Pulp Demonstrate Different Chondrogenic Efficiency in the 3D Spheroid Culture” described the chondrogenesis capability of mesenchymal stromal cells derived from different sources in 3D spheroid culture ex vivo. The topic is not novel, but the authors are off to a good start by presenting a thorough analysis to compare directly the immunophenotype and chondrogenic capabilities of MSCs differentiated in current spheroid platform. There are some issues that need to be addressed in the manuscript:
- What’s the major difference between this manuscript and previously published study (Ref 12)? Only the control group and donors were different. The same MSC sources and methods were studied from the same team.
- MSCs from different anatomic locations have differentiation potentials and differing requirements for, and responses to inductive stimuli, making it necessary to optimize culture conditions and differentiation protocols for each cell source. TGF-β induction has been used widely in MSCs, but some additional factors may be needed for chondrogenesis between different donor sites, such as bone morphogenetic protein-6 could promote ATMSCs in chondrogenesis. Similarly, MSCs derived from dental pulp and umbilical cord vein may react differently under standard induction conditions. Testing different induction reagents or adding a control group (no TGF-β added) would be helpful if only in vitro study is planned.
- What’s the passage of MSCs in spheroid formation?
- As the authors presented in Figure 3, osteogenic and adipogenic differentiation capacity of those MSCs was proven in 2D culture. How about the chondrogenic differentiation in 2D culture?
- In page 5, “Histochemical analysis demonstrated that by day 21 of culture in chondrogenic differentiation medium, ECM accumulation was observed in all spheroids, being especially 140 pronounced in the case of chondrocytes (Fig. 4)” and “no significant ECM accumulation was evident in spheroids formed by SHED or WJMSCs.” Is the histochemical analysis performed here a summary of the masson and alcian blue staining? The first two paragraphs in the section were confusing. Please describe the results of day 14 and day 21 clearly. Grossly, the GAG content was increased in all the groups from day 14 to day 21. The collagen content (stained by masson) was obviously increased overtime in AT, WJ and chondrocyte groups too but in different arrangement and zones. What are the possible causes of different ECM deposition in zones? Nutrition permeability has an effect on MSC differentiation?
- Figure labelling. In “2.6. Changes in the cartilage-specific gene expression” section, it is Figure 8 instead of Figure 6 that was discussed. The X axis labeling was confusing. Please add the proper text to reflect the dates and groups.
- One of the downsides of using SHED is the limited tissue volume, even though the chondrogenic capability has been proven in many studies. The cartilage repair for OA in adults also is less applicable with WJMSCs when autologous tissue/cells, such as ATMSCs derived from fat, are available and usually in large supply. WJMSCs could be a better option for ear reconstruction in children since ATMSCs is limited in this condition.
- In vivo study in cartilage repair models would be a plus.
Author Response
Dear Editors and Reviewers,
Thank you for reviewing our manuscript, ijms-2817617, entitled «Mesenchymal Stromal Cells Isolated from the Umbilical Cord, Adipose Tissue and the Deciduous Tooth Pulp Demonstrate Different Chondrogenic Efficiency in the 3D Spheroid Culture» and valuable comments and suggestions. We tried to amend the manuscript accordingly to comply with Journal’s requirements.
Please, find our point-by-point responses to the reviewers’ comments below.
Answers to Reviewer 1.
Question 1.
What’s the major difference between this manuscript and previously published study (Ref 12)? Only the control group and donors were different. The same MSC sources and methods were studied from the same team.
Answer 1.
This article can be seen as the continuation of our previous work. However, now we present the results of a much more complex study involving several new methods such as qPCR and biochemical analysis of GAG/DNA content. The previous article lacked quantitative measurements, presented in the submitted manuscript. We also improved the quality of the histological data and believe that it enhanced the overall quality of the study. We feel that our data presented in the manuscript has a significant impact in the field and provided the following novel insights into studied problem of articular cartilage regeneration.
Question 2.
MSCs from different anatomic locations have differentiation potentials and differing requirements for, and responses to inductive stimuli, making it necessary to optimize culture conditions and differentiation protocols for each cell source. TGF-β induction has been used widely in MSCs, but some additional factors may be needed for chondrogenesis between different donor sites, such as bone morphogenetic protein-6 could promote ATMSCs in chondrogenesis. Similarly, MSCs derived from dental pulp and umbilical cord vein may react differently under standard induction conditions. Testing different induction reagents or adding a control group (no TGF-β added) would be helpful if only in vitro study is planned.
Answer 2.
It is true that different inducers can be used as chondrogenic stimuli during the in vitro chondrogenic differentiation of MSCs. However, the focus of the present work was limited to comparing cells from different sources under exactly the same conditions. TGF-beta1 was chosen as the most effective inducer based on the literature and our own studies. We find the topic very interesting and plan to compare different factors in the future.
As for the control group, we first tried to culture the spheroids in the growth medium, but in the absence of differentiation induction, the proliferation of the cells inside the spheroids did not slow down, leading to spheroid overgrowth, formation of necrotic cores and finally disaggregation on days 10-14. The data was not considered worthy of inclusion.
Question 3.
What’s the passage of MSCs in spheroid formation?
Answer 3.
The following text has been included in section 4.5:
Cell cultures at the passages 4-6 were used for the experiments.
Question 4.
As the authors presented in Figure 3, osteogenic and adipogenic differentiation capacity of those MSCs was proven in 2D culture. How about the chondrogenic differentiation in 2D culture?
Answer 4.
Optimal conditions for in vitro chondrogenic differentiation involve three-dimensional aggregation of cells into spheroids, where high cell density and cell-cell interactions play an important role in the mechanism of chondrogenesis. We believe that 3D spheroid culture provides more physiologically relevant model for studying chondrogenic differentiation.
Question 5.
In page 5, “Histochemical analysis demonstrated that by day 21 of culture in chondrogenic differentiation medium, ECM accumulation was observed in all spheroids, being especially 140 pronounced in the case of chondrocytes (Fig. 4)” and “no significant ECM accumulation was evident in spheroids formed by SHED or WJMSCs.” Is the histochemical analysis performed here a summary of the masson and alcian blue staining? The first two paragraphs in the section were confusing. Please describe the results of day 14 and day 21 clearly. Grossly, the GAG content was increased in all the groups from day 14 to day 21. The collagen content (stained by masson) was obviously increased overtime in AT, WJ and chondrocyte groups too but in different arrangement and zones. What are the possible causes of different ECM deposition in zones? Nutrition permeability has an effect on MSC differentiation?
Answer 5
We agree with the reviewer and have changed the description of the results of the 14th and 21st days to a more understandable one:
Histochemical analysis demonstrated that ECM accumulated in all types of spheroids (Fig. 4), but its amount and composition differed among them. Collagen in ECM was detected by Masson staining; Alcian blue staining was applied to detect GAG synthesis as an indicator of cells undergoing chondrogenic differentiation. On day 14, the highest accumulation of ECM was observed in chondrocyte spheroids. Among MSC spheroids, ECM accretion was greatest in the ATMSC spheroids. By day 14 collagen and GAG were visualized in all spheroid types, with a higher cumulation in chondrocyte spheroids.
Results obtained on day 21 of culturing spheroids in chondrogenic differentiation medium are of particular interest. ECM accumulation was found to be especially pronounced in chondrocyte spheroids. In these spheroids, specific areas were observed, which had a higher ratio of ECM to cells. Chondrocytes in these areas were not tightly surrounded by matrix, which is characteristic of the juvenile chondrocytes. Considerable ECM accumulation also occurred in ATMSC spheroids. On the contrary, no significant ECM accumulation was in spheroids formed by SHED or WJMSCs.
For all spheroid types, the collagen production in ECM was higher on day 21. Masson staining for collagen showed the strongest positive reaction in ATMSC-derived spheroids. Collagen fibers were notably visible at the periphery of chondrocyte and ATMSC spheroids, as well as in the central region of WJMSC spheroids. In SHED spheroids, the collagen staining was of low intensity.
The GAG content in all spheroids increased over time of culture, and its highest level was observed in chondrocyte spheroids on day 21. Among MSC spheroids, the ECM in ATMSC and WJMSC spheroids was stained more intensely than in SHED spheroids.
Thus, when cultured in chondrogenic differentiation medium, all spheroids showed GAG accumulation increasing from day 14 to day 21. Radial heterogeneity of collagen distribution can be associated with inefficient diffusion of nutrients and differentiation inducers inside the spheroids.
Question 6.
Figure labelling. In “2.6. Changes in the cartilage-specific gene expression” section, it is Figure 8 instead of Figure 6 that was discussed. The X axis labeling was confusing. Please add the proper text to reflect the dates and groups.
Answer 6.
Due to a technical error, we mislabeled the X-axis in the original manuscript. We apologize for the confusion. Proper labelling and text corrections have been added.
Question 7.
One of the downsides of using SHED is the limited tissue volume, even though the chondrogenic capability has been proven in many studies. The cartilage repair for OA in adults also is less applicable with WJMSCs when autologous tissue/cells, such as ATMSCs derived from fat, are available and usually in large supply. WJMSCs could be a better option for ear reconstruction in children since ATMSCs is limited in this condition.
Answer 7.
Depending on their tissue source, every type of MSCs has certain advantages and disadvantages. Actually, that is why we studied different types of these cells. SHEDs, despite limited initial tissue volume, can give sufficient quantities of cells for clinical applications. We fully agree with your comments on the practical applicability of different types of MSCs. However, our goal was to evaluate some of the types as potential starting material for the joint cartilage reconstruction.
Question 8.
In vivo study in cartilage repair models would be a plus.
Answer 8.
We absolutely agree. We plan to conduct an in vivo study in the near future using a rabbit OA model.

Reviewer 2 Report
Comments and Suggestions for Authors
The authors conclude that ATMSCs, when cultured as spheroids, have the highest chondrogenic potential and WJMSCs also appear to be a promising cell source for cartilage tissue engineering. Therefore, the results suggest the feasibility of Wharton’s jelly as a valuable cell source for cartilage engineering applications as well as for osteoarthritis treatment.
This manuscript refers to useful information and is reasonable for the authors’ hypothesis. However, I think there are some unclear and insufficient points.
Major comments
1. Figure 2/3: The authors should add the chondrocyte data (negative data).
2. Figure 5: 1) The day of this data/samples was not indicated. I can guess this day is 14d or 21d, but the authors should indicate this. Moreover, the authors had better show the other data which means the authors should show both data od 14d and 21d.
2) I recommend that the author add each quantitative data, for example, counting each positive cell.
Minor comments
1. Figure 2: The authors should organize each Y-axis scale to compare the expression intensity among them.
2. Figure 3/5: The authors should change to the lower magnified photos.
3. Figure 4: These pellet sizes are different; however, the authors have not mentioned the reason. Please mention this difference in the DISCUSSION section.
4. Figure 6A: Compared to the data of Figure 4, each DNA amount does not fit each pellet size. I think the authors should mention the reason in the DISCUSSION section.
5. Figure 8: This result section (#2.6) does not match the Figure 8. Please check      again and revise this. Moreover, there is no explanation for #1 to 12 on      the X-Axis. Please explain them in the figure legend or RESULT section.      and revise their indication.
6. DISCUSSION: The authors have referred to immunosuppressive and anti-inflammatory activity in the DISCUSSION section. However, there is no data for such activity of MSCs in this manuscript. If the authors performed some experiments for them, please add them as supplemental data.
Author Response
Dear Editors and Reviewers,
Thank you for reviewing our manuscript, ijms-2817617, entitled «Mesenchymal Stromal Cells Isolated from the Umbilical Cord, Adipose Tissue and the Deciduous Tooth Pulp Demonstrate Different Chondrogenic Efficiency in the 3D Spheroid Culture» and valuable comments and suggestions. We tried to amend the manuscript accordingly to comply with Journal’s requirements.
Please, find our point-by-point responses to the reviewers’ comments below.
Answers to Reviewer 2.
Major comment 1.
Figure 2/3: The authors should add the chondrocyte data (negative data).
Answer.
Figures 2 and 3 refer specifically to the characterization of multipotent mesenchymal stromal cells as per ISCT criteria, therefore mature primary chondrocytes were not included in this part of the study. Following your recommendation, we added negative data for native (undifferentiated) cells to figure 3.
Major comment 2.
Figure 5: 1) The day of this data/samples was not indicated. I can guess this day is 14d or 21d, but the authors should indicate this. Moreover, the authors had better show the other data which means the authors should show both data od 14d and 21d. 2) I recommend that the author add each quantitative data, for example, counting each positive cell.
Answer.
1) We agree. The day of data/samples has been indicated.
2) Unfortunately, it is not possible to make a quantitative assessment by counting cells on the presented photographs, as the staining is mainly visualized in the ECM. However, we used qPCR and GAG/DNA content measurements to quantitatively evaluate chondrogenic differentiation.
Minor comment 1.
Figure 2: The authors should organize each Y-axis scale to compare the expression intensity among them.
Answer.
In Figure 2, we used the most common way to present flow cytometry data as histograms. The X axis shows the fluorescence intensity (marker expression level), and the Y axis shows the number of events (analyzed cells). All X axes have the same scale. We placed all histograms with the same marker in a separate column, which simplifies the comparison of the marker expression in different types of MSCs. The differences in the Y axis are due to some variation in the number of cells analyzed in the samples. Since these differences are small and do not affect the estimation of fluorescence intensity values, we did not present normalized values for this parameter.
Minor comment 2.
Figure 3/5: The authors should change to the lower magnified photos.
Answer.
Unfortunately, we don't have the technical means to reduce the magnification of the photos in Figure 3, but we have added the photos of the whole spheroids in Figure 5, as you suggested.
Minor comment 3.
Figure 4: These pellet sizes are different; however, the authors have not mentioned the reason. Please mention this difference in the DISCUSSION section.
Answer.
The following text was added to part 2.2:
The resulting SHED spheroids were visually smaller than spheroids built of the other cell types, probably because the cells themselves were smaller.
Minor comment 4.
Figure 6A: Compared to the data of Figure 4, each DNA amount does not fit each pellet size. I think the authors should mention the reason in the DISCUSSION section.
Answer.
The following text was added to part 2.5:
Note, that the amount of DNA and the diameter of the spheroids do not always conform (Fig. 4). This is because the size of the spheroid depends not only on the number of cells, but also on the amount of ECM accumulated during chondrogenic differentiation.
Minor comment 5.
Figure 8: This result section (#2.6) does not match the Figure 8. Please check again and revise this. Moreover, there is no explanation for #1 to 12 on the X-Axis. Please explain them in the figure legend or RESULT section and revise their indication.
Answer.
Due to a technical error, we mislabeled the X-axis in the original manuscript. We apologize for the confusion. Proper labelling and text corrections have been added.
Minor comment 6.
DISCUSSION: The authors have referred to immunosuppressive and anti-inflammatory activity in the DISCUSSION section. However, there is no data for such activity of MSCs in this manuscript. If the authors performed some experiments for them, please add them as supplemental data.
Answer.
Actually, in our research we have obtained some data regarding the immunomodulatory activity of MSCs isolated from various sources (doi: 10.1007/s10517-017-3658-5; 10.46582/jsrm.1302009.eCollection2017; 10.3390/biomedicines9101286; 10.47056/1814-3490-2023-4-249-256). The data include a comparison of the immunological properties of MSCs and NSCs, as well as MSCs from normal tissue and pathological lesion. These data were obtained using 2D cultivation and have already been published. This is why we cannot add them as a supplement. In the future, we plan to use co-culture spheroids to study the immunological properties of MSCs.

Round 2
Reviewer 1 Report
Comments and Suggestions for Authors
None.
Reviewer 2 Report
Comments and Suggestions for Authors
I thank the authors for your answers and revisions.
I checked these answers and revisions, and I have agreed with all your answers/revisions.
I think this revised manuscript can be acceptable.